# The Relationship between Endophytic Fungi of *Chimonanthus praecox* and Volatile Metabolites under Different Circadian Rhythms and Blooming Stages

**DOI:** 10.3390/jof10020145

**Published:** 2024-02-11

**Authors:** Yue Li, Jingying Hei, Xiahong He, Rui Rui, Shu Wang

**Affiliations:** 1Key Laboratory of Underforest Resource Protection and Utilization in Yunnan Province, College of Landscape and Horticulture of Southwest Forestry University, Kunming 650224, China; liyue_421@163.com; 2Key Laboratory of Ministry of Education on Forest Resources Conservation and Utilization in Southwest Mountainous Area, Kunming International Research and Development Center of Ecological Forestry Industry, Kunming 650233, China; h2940794807@163.com (J.H.); hxh@swfu.edu.cn (X.H.)

**Keywords:** *Chimonanthus praecox*, endophytic fungi, floral fragrance, diversity, community composition

## Abstract

*Chimonanthus praecox* is an aromatic plant that flowers in winter. The composition of the floral volatiles of *C. praecox* is influenced by different blooming stages, circadian rhythms and species. However, the relationship between floral volatiles and plant endophytic fungi has not received much research attention. Here, we used high-throughput sequencing technology to compare and analyze the changes in the structure and diversity of the endophytic fungal communities in *C. praecox* under different circadian rhythms (7:00 a.m., 1:00 p.m., and 7:00 p.m.) and in different blooming stages (unopened flowers and opened flowers). The endophytic fungi of *C. praecox* consisted of nine phyla, 34 classes, 79 orders, 181 families, 293 genera, and 397 species, and Ascomycota was the dominant phylum. Under a diurnal rhythm, the diversity (Chao1 and Shannon indices) of endophytic fungi gradually decreased in the unopened flowers, while an increasing and then decreasing trend was found for the opened flowers. In the different blooming stages, the endophytic fungal diversity was significantly higher at 7:00 a.m. in the unopened flowers compared to the opened flowers. Humidity was the key factors that significantly affected the endophytic fungal diversity and community. Moreover, 11 endophytic fungi were significantly positively or negatively correlated with seven floral volatiles. In conclusion, the community structure and diversity of endophytic fungi in *C. praecox* were affected by the different blooming stages and circadian rhythms, and a correlation effect related to floral volatiles was found, but there are other possible reasons that were not tested. This study provides a theoretical basis for elucidating the interrelationships between endophytic fungi, floral volatiles, and environmental factors in *C. praecox*.

## 1. Introduction

Plant endophytes, mainly including fungi, bacteria, and actinomycetes, commonly inhabit healthy plant tissues without causing apparent disease [1]. Endophytic fungi can promote plant growth [2], enhance tissue colonization [3], improve stress resistance [4], and produce bioactive secondary metabolites [5,6]. They can synthesize the same or similar compounds as their hosts, and increase key metabolite production in horticultural plants, such as paclitaxel (*Taxus chinensis*) [7], methyl eugenol (*Rosa* spp.) [5], and 4-penten-2-ol (*Lilium brownii*) [8]. Similar phenomena occur in medicinal plants. For instance, the endophytes in *Azadirachta indica* [9] and *Rhodiola sachalinensis* [10] can produce the AL4 and rhodioside metabolites. These results suggest that endophytic fungi and their hosts share secondary metabolic pathways or production sources. Therefore, revealing the close relationship between endophytic fungi and metabolites can clarify the sources and metabolic pathways of plant metabolites.

The community composition, abundance, and diversity of plant endophytic fungi are primarily influenced by the host species [11,12], growth period [13], and geographic habitat [14], among other factors. Taxa of endophytic fungi significantly differ in different ornamental plant species: for example, *Rhododendron* spp. [11], *Paeonia ludlowii* [15], and *Lonicera japonica* [16] were found to contain 43, 14, and 19 genera, respectively. Similar results were observed in different floral organs for instance, 11 and 4 genera were found in Eucalyptus globulus [17] and *Moinga oleifera* flowers [18], respectively. Moreover, endophytic fungal diversity and community composition vary between different plant tissues (roots, stems, and leaves) across growth stages. For instance, endophytic fungal diversity was higher in *Bletilla mycorrhizae* in the flowering and emergence periods than in other periods [19]. The changes in the major genera (*Penicillium, Fusarium, Xylaria*) in *Capsicum annuum* stems across the seedling, flowering, and fruiting stages were inconsistent [20]. Similar variability was found in the leaves of *Stellera chamaejasme* [21]. In addition, environmental factors such as temperature and humidity have significant impacts on the diversity of endophytes. Previous results showed that endophytic fungal diversity in angiosperms is richer in tropical versus temperate/cold forests [22]. Humid climates were found to be more suitable for the survival of endophytes than arid environments [23].

Endophytic fungi can not only produce secondary metabolites in plant tissue, but are also one of the critical pathways for producing volatile compounds with a floral fragrance in plants [24]. Five aromatic metabolites (cyclo-(Trp-Ala), indole-3-carboxylic acid (ICA), in-dole-3-carbaldehyde, mellein, and 2-phenylethanol) were identified in endophytic fungi of *Viscum coloratum* flowers [25], confirming that endophytic fungi in flowers may be an essential resource for natural plant aromas. Moreover, the emission and composition of floral volatile compounds in different plants are affected by the blooming stage [26,27,28,29], circadian rhythms [28,30], environmental factors [31,32,33,34], and species [35]. However, it is unclear whether floral substance changes caused by biological and abiotic factors can affect some endophytic fungal populations. At the same time, environmental factors can directly affect not only floral volatiles, but also the diversity of endophytic fungi. Therefore, exploring the relationship between floral volatiles and endophytic fungi under different conditions is a necessary way to clarify the source and production of floral volatiles.

*Chimonanthus praecox* is a deciduous shrub that flowering in winter, belonging to the genus *Chimonanthus* in the family Calycanthaceae, and it is a fragrant and unique traditional plant in China. Most previous studies showed that the components of floral volatiles in *C. praecox* are altered by the blooming stage [35,36], circadian rhythms [37], geographic locations [38], and species [39]. Although a close relationship between endophytic fungi and floral volatiles has been demonstrated, the relationship between endophytic fungi diversity, floral volatiles, and environmental factors in *C. praecox* is still unclear. Based on our previous study of the variation in the floral volatiles of C. praecox in different blooming stages (unopened flowers and opened flowers) and under different circadian rhythms (7:00 a.m., 1:00 p.m., and 7:00 p.m.) in Kunming, Yunnan Province [37], we further analyzed the community structure and diversity of the endophytic fungi in *C. praecox* using high-throughput sequencing technology. This study aimed to (1) investigate the influence of blooming stages and circadian rhythms on the community structure and diversity of the endophytic fungi in *C. praecox*, and (2) elucidate the relationship between endophytic fungal diversity, floral volatiles, and environmental factors in *C. praecox*.

## 2. Materials and Methods

### 2.1. Plant Materials

In this experiment, the flowers of *var. grandiflorus* in different blooming stages were used as research materials. The unopened flowers were yellow with upright stamens and buds beginning to increase, while the opened flowers were yellow with completely opened petals and a purple flower center (Figure 1). All the flowers were collected from the campus of Southwest Forestry University in Kunming, Yunnan Province.

### 2.2. Flower Samples Collection

On 25 December 2019, 24 unopened and 18 opened flowers of *C. praecox* were collected at 7:00 a.m., 1:00 p.m., and 7:00 p.m.. Two opened flowers and four unopened flowers (about 0.5 g) were placed in a 20 mL head space bottle, which was quickly sealed with an aluminum cover, to determine the floral volatiles. Additionally, twelve unopened and opened flowers of *C. praecox* were selected and placed in sterile self-sealing bags, and then immediately brought back to the laboratory for surface disinfection and biocidal treatment. The disinfection process involved the following steps: 75% alcohol for 60 s, 0.5% sodium hypochlorite for 90 s, and 75% alcohol for 30 s. After each step finished, sterile excess water was applied 3–4 times to rinse the samples. Subsequently, all the flower samples were dried with sterile filter paper, placed in a 50 mL sterile centrifugal tube, and stored at −80 °C for DNA extraction and sequencing.

### 2.3. Floral Volatile Collection and Identification

Headspace solid-phase microextraction (HS-SPME) and gas chromatography–mass spectrometry (GC/MS) were carried out to extract the floral scent compounds in *C. praecox*. An empty capped vial was used as the negative control. The SPME device was equipped with an SPME fiber (50/30 μm) coated with polydimethylsiloxane (PDMS) (TriPlus 300, Thermo Fisher Scientific, Waltham, MA, USA). In detail, the protocols for HS-SPME followed the methods of Chen et al. [40]. Subsequently, a GC/MS system (Trace GC Ultra/ITQ900, Thermo Fisher Scientific, Waltham, MA, USA) with an HP-5MS capillary column (30 m × 0.25 mm × 0.25 μm, Agilent J & W Scientific, Santa Clara, CA, USA) was performed to identify the floral volatiles. The variation in the GC oven temperature and the data recorded from the mass spectrometer in electron impact mode (MS/EI) were based on the methods of Li et al. [37].

### 2.4. DNA Extraction, PCR Amplification, and Sequencing

DNA was extracted from 0.5 g of frozen flowers using the FastDNA^®^ Spin KitDNA (MP Biomedicals, Southern California, USA) kit according to the manufacturer’s protocol. The obtained DNA was quantified by using a NanoDrop^®^ spectrophotometer (ND-2000c, Thermo Scientific, Waltham, MA, USA) for nucleic acid concentration and purity. The entire fungal ITS1 region was amplified using the primers ITS1F (5′-TCC GTA GGT GAA CCT GCG G-3′) and ITS2R (5′-TCC TCC GCT TAT TGA TAT GC-3′) [41]. The amplification system consisted of 5 × Fast Pfu Buffer 4 µL, 2.5 mM dNTPs 2 µL, Forward Primer (5 µM) 0.8 µL, Reverse Primer (5 µM) 0.8 µL, Fast Pfu Polymerase 0.4 µL, BSA 0.2 µL, Template DNA 10 ng, and ddH_2_O added to 20 µL. The PCR protocols were as follows: pre-denaturation at 95 °C for 3 min; denaturation at 95 °C for 20 s, annealing at 55 °C for 30 s, and extension at 72 °C for 30 s, over 30 cycles; and extension at 72 °C for 5 min. Three PCR products as three replicates were selected to detect the DNA quality using 2% agarose gel electrophoresis. The qualified PCR products were purified with an AxyPrep DNA Gel Extraction Kit and quantified using a Quantus™ Fluorometer. Amplicons from different samples were mixed for library construction using a NEXTFLEX Rapid DNA-Seq Kit. Finally, high-throughput sequencing was performed using the MiSeq PE300 platform (Shanghai MajorBio Technology Co, China). The sequences have been deposited in the National Center for Biotechnology Information Sequence Read Archive under Accession No. PRJNA875038 (ITS data).

### 2.5. Data Processing and Statistical Analysis

In this experiment, microbiome bioinformatics was implemented in QIIME 1.9.1, with slight modifications according to the official tutorials (www.majorbio.com, accessed on 19 November 2022). Briefly, raw sequence data were quality-controlled using the Trimmomatic plugin, followed by base splicing using the FLASH plugin. Finally, the sequences were quality-frequency-analyzed, denoised, and merged, and chimeras were removed using the UCHIME plugin. The RDP classifier was used to annotate the species classification for each sequence, and representative sequences were assigned by comparing to Silva database (ssu123), with the comparison threshold being 70%. Sequence data analysis was performed using QIIME 1.9.1, during which non-fungal sequences were removed to avoid contamination of the chloroplasts and mitochondria of the host plant.

The mothur plugin was employed to calculate the alpha diversity (Chao1 and Shannon indices). PCoA plots based on the Bray–Curtis index were analyzed using the vegan and ggplot2 packages in R (version 3.3.1). Venn diagrams based on an OTU table with a 97% similarity level and a heat map of the correlation clustering for the top 30 genera and 10 main floral volatiles (our pre-study) were generated using R (version 3.3.1). RDA analysis was performed using the vegan package in R to analyze the relationship between endophytic fungal communities and environmental factors.

## 3. Results

### 3.1. Alpha and Beta Diversity of Endophytic Fungi

The detailed sequencing information of the endophytic fungi in *C. praecox* in different blooming stages and under different circadian rhythms can be found in the attached table (Appendix A). The rarefaction analysis results (Appendix A) indicated an adequate sequencing depth for all samples. The α diversity (Chao1 and Shannon indices) of the endophytic fungal community in *C. praecox* was analyzed using Qiime software (1.9.1) (Figure 2a,b). Our findings indicated a circadian rhythm-dependent variation in these indices. Specifically, endophytic fungal α diversity showed a gradual decrease in the unopened flowers with the circadian rhythm, whereas an initial increase followed by a decrease was found in the opened flowers. In the different blooming stages, the Chao1 and Shannon indices of the endophytic fungi were significantly higher at 7:00 a.m. in the unopened flowers compared to the opened flowers (*p* < 0.05). Additionally, the community abundance and diversity of the endophytic fungi at the other two time points (1:00 p.m. and 7:00 p.m.) showed a no differences between the unopened and opened flowers. The beta diversity of endophytic fungi was further analyzed using the Bray–Curtis distance measure. The PCoA plot (Figure 2c) showed that the contributions of PC1 and PC2 were 32.3% and 13.29%, respectively, with no significant differences between the groups (R^2^ = 0.22, *p* > 0.05).

### 3.2. Taxonomic Analysis of Endophytic Fungal Communities

Following clustering at the 97% similarity level, the Venn diagram (Figure 3a) revealed that the number of Operational Taxonomic Units (OTUs) of the endophytic fungi in the unopened flowers of *C. praecox* decreased gradually with the circadian rhythm, while in the opened flowers, an initial increase followed by a decrease was observed. The GUA, GUB, GUC, GOA, GOB, and GOC samples contained 89, 65, 29, 34, 52, and 35 OTUs, respectively, with a total of 32 shared OTUs. Among these, GUA had the highest number of OTUs, with 252. The top three most abundant endophytic fungi were unclassified Ascomycota (35.37%), *Didymella* (13.63%), and *Apiotrichum* (6.54%). The Venn diagrams for the different blooming stages (Figure 3b) showed that the number of OTUs in the unopened flowers (202) was higher than in the opened flowers (130), with a total of 165 OTUs for both stages. The top three endophytic fungi were unclassified Ascomycota (30.40%), *Didymella* (11.49%), and *Apiotrichum* (5.85%).

OTU representative sequences with 97% or higher similarity levels were taxonomically analyzed using the RDP classifier Bayesian algorithm. The *C. praecox* sample contained various endophytic fungal species types, including nine phyla, 34 classes, 79 orders, 181 families, 293 genera, and 397 species. The top five endophytic fungal communities were identified at the phylum, class, order, family, genus, and species levels (Table 1). Ascomycota, Dothideomycetes, and Pleosporales were the most dominant phylum, class, and order, respectively, while fungi belonging to unclassified Ascomycota were the most prevalent at the family, genus, and species levels. The percentages of Ascomycota and Dothideomycetes in the unopened and opened flower stages decreased with the circadian rhythm and were the highest at 7:00 a.m. Similar results were observed in the most dominant order (Pleosporales) and family (unclassified Ascomycota). At the genus and species levels, the percentage of unclassified Ascomycota decreased first and then increased with the circadian rhythm, and was the highest at 7:00 a.m. in the unopened and opened flower stages.

### 3.3. Correlation Analysis between Endophytic Fungi and Floral Volatiles in C. praecox

A classification analysis was conducted at the 97% similarity level using the Unite database for 293 genera, and then the top 30 genera in terms of relative abundance were selected for the correlation analysis (Appendix A). *Didymella* was the dominant genus in unopened and opened flowers. The abundance of Didymella was increased in the unopened flowers and decreased in the opened flowers with the circadian rhythm, respectively. Additionally, the correlation analysis between the top 30 main genera of endophytic fungi and 10 main floral compounds showed that only 11 fungi were significantly associated with seven floral volatiles (Appendix A, Figure 4). Unclassified Saccharomycetales (R = −0.56; R = −0.75), *Saccharomycosis* (R = −0.59; R = −0.67), *Wickerhamomyces* (R = −0.56; R = −0.64), and *Cercospora* (R = −0.56; R = −0.65) were negatively correlated with M-Xylene and o-toluic acid, and positively correlated with *Diaporthe* (R = 0.55; R = 0.54). Meanwhile, *Saccharomyces* (R = −0.56) and unclassified Ascomycota (R = 0.58) were associated with o-toluic acid, respectively. In addition, *Rhodotorula* was negatively correlated with Cyclohepta-1,3,5-triene (R = −0.58), unclassified Necritiaceae and Eugenol (R = −0.49), unclassified Cystobasidiomycetes and Benzyl alcohol (R = −0.57) and (-)-Myrtenol (R = −0.54), and *Gibberella* was significantly positively correlated with Germacrene D (R = 0.50). Furthermore, the community composition and diversity of endophytic fungi in *C. praecox* was positively correlated with humidity (Appendix A; Appendix A).

## 4. Discussion

### 4.1. Endophytic Fungal Diversity in C. praecox Is Altered by Circadian Rhythm and Blooming Stage

The diversity and community structure of endophytic fungi are predominantly determined by the host species, growth stages, and environmental conditions [12,13,14]. Our study indicated a circadian rhythm-dependent variation in the Chao1 and Shannon indices of the endophytic fungi in *C. praecox*. Specifically, these indices exhibited a gradual decrease in the unopened flowers, while in the opened flowers, an initial increase followed by a decrease was observed. Despite the fact that limited research is available about the impact of circadian rhythms on plant endophytic fungi, it has been established that the microbial flora exhibits heightened sensitivity to temperature fluctuations, particularly over short durations such as daily variations [42]. However, the correlation analysis revealed a positive correlation between the diversity and community structure of *C. praecox*’s endophytic fungi and humidity. This aligns with findings from studies on *Taxus Chinensis* [43], and can be attributed to the organism’s preference for dark and humid environments. Given these observations, we propose that the diversity of endophytic fungi in *C. praecox* is modulated by humidity, which is, in turn, influenced by circadian rhythms. Additionally, we observed inconsistent variations in endophytic fungal diversity in different blooming stages (unopened and opened flowers), a phenomenon also reported in other studies [19,20,21,44]. It is known that endophytic fungi in different growth stages have distinct nutrient requirements, which could significantly influence their behavior during the flowering process [45]. Our study also suggested that the β diversity of the endophytic fungi remained relatively consistent across the different blooming stages and circadian rhythms. This could imply that endophytic fungi are not easily selected, being subjected to environmental filtering over short periods or throughout growth [46], especially within the same plant. In addition, the endophytic fungi in *C. praecox* are influenced by both circadian rhythms and blooming stages, similar to the dynamic changes observed in floral volatiles [47,48]. As insect pollinators act rhythmically, the variations in floral scent with rhythm are associated with the activity of the corresponding pollinators [37]. This leads us to speculate about the potential close relationship between plant endophytic fungi and their secondary metabolites. In other words, changes in floral volatiles may be an important reason for changes in endophytic fungi.

### 4.2. Dominant Fungal Communities under Different Circadian Rhythms and in Different Blooming Stages

We analyzed the species composition and relative abundance of endophytic fungi in *C. praecox*. Our analysis revealed that the relative abundance of endophytic fungi in *C. praecox* at various taxonomic levels fluctuated with the circadian rhythm. Specifically, the number of endophytic fungal OTUs decreased gradually in the unopened flowers, whereas an initial increase followed by a decrease was observed with the circadian rhythm in the opened flowers. Moreover, the species composition of endophytic fungi in *C. praecox* spans nine phyla, 34 classes, 79 orders, 181 families, 293 genera, and 397 species. Notably, Ascomycota was dominant in *C. praecox* and prevalent in different tissue parts of other plants, including the seeds, leaves, roots, and bark [49,50,51,52,53]. Aromatic compounds produced by Ascomycota through biotransformation pathways [54] can be used as food by some endophytic fungi [55], and these endophytic fungi can also resist the toxicity of the compounds [56], which could potentially account for their high presence in *C. praecox*. However, this finding contrasts with the primary fungal communities of local flowers in Yunnan, which predominantly belong to the Deuteromycotina, suggesting a potential influence of the host species [57]. In addition to Ascomycota, *Didymella* holds a dominant position at the genus level among the endophytic fungi of *C. praecox*. As the most phytopathologically important genus in the Didymellaceae, *Didymella* may serve as a significant reservoir of species of the Septosporaceae in plants [58]. Furthermore, the endophytic fungi of *C. praecox* consisted of Phaeosphaeriaceae, *Aspergillus*, and *Gibberella*. Among these, some species within the Phaeosphaeriaceae have been identified as a source of polyketide biosynthesis in desert plants [59]; *Aspergillus* synthesizes a variety of secondary metabolites [60]; and *Gibberella* has been reported to promote plant growth [61]. To summarize, our findings are expected to enrich the understanding of the types and resources of endophytic fungi in *C. praecox*.

### 4.3. Relationship between Endophytic Fungi and Floral Volatiles in C. praecox

Eleven endophytic fungi were positively or negatively correlated with seven floral volatiles in *C. praecox*. This may be attributed to the ability of endophytic fungi to be directly selected from within the plant, thereby influencing its metabolic pathways [62]. Furthermore, some aromatic hydrocarbons (e.g., α-Terpinene, M-xylene, O-toluic acid, and Germacrene D) present in the floral volatiles of *C. praecox* exhibited a positive correlation with specific endophytic fungi (Ascomycota, *Gibberella*, and *Diaporthe*). This contrasts with previous findings that phenolic substances in medicinal plants are more likely to coexist with certain endophytic fungi, possibly due to variations in the host species or metabolic responses [63]. The dominance of aromatic hydrocarbons in *C. praecox*, an ornamental flower, suggests a different metabolic preference compared to medicinal plants, where phenolic substances enhance the antioxidant capacity. *Diaporthe* fungi, a significant source of functional natural products, have been found to produce polyketones, alkaloids, terpenoids, anthraquinones, and other novel structural metabolites [64]. Similarly, most *Gibberella* fungi are known to produce bioactive metabolites [65]. Interestingly, M-xylene and O-toluic acid showed significant negative correlations (*p* < 0.01) with unclassified *Saccharomycetales*, *Saccharomycopsis*, and *Wickerhamomyces*. This parallels findings of negative correlations between endophytic fungi and some precursors for the synthesis of paclitaxel in *Taxus Chinensis* [66], implying a close relationship between endophytic fungi and plant metabolism. In conclusion, the endophytic fungi of *C. praecox* can directly affect the production of secondary metabolites, which may be influenced by multiple strains of fungi. This study further deepens our understanding of the relationship between plant endophytic fungi and their floral volatiles.

## 5. Conclusions

This study represents the first investigation into the community structure and diversity of endophytic fungi in relation to the circadian rhythms and blooming stages of *C. praecox* using high-throughput sequencing technology. Under a diurnal rhythm, the diversity (Chao1 and Shannon indices) of endophytic fungi gradually decreased in the unopened flowers, while it showed an increasing and then decreasing trend in the opened flowers. In the different blooming stages, the endophytic fungal diversity was significantly higher at 7:00 a.m. in the unopened flowers compared to the opened flowers. Ascomycota was found to be dominant in *C. praecox* flowers. Humidity was the key factors that significantly affected the endophytic fungal diversity and community. Notably, 11 endophytic fungi were significantly positively or negatively correlated with seven floral volatiles. Our findings are expected to enrich the knowledge of the types and resources of endophytic fungi in *C. praecox* and further elucidate the relationship between endophytic fungi and floral volatiles.

## Figures and Tables

**Figure 1 jof-10-00145-f001:**
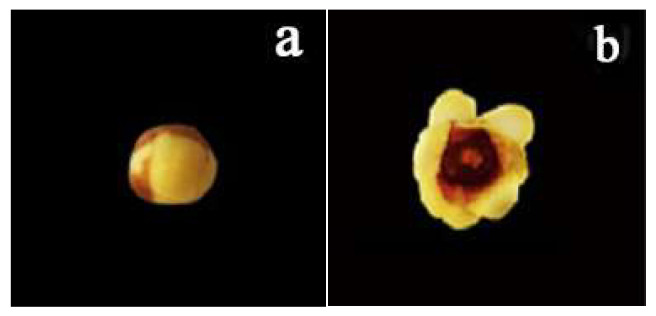
Flower morphology of *var. grandiflorus* ((**a**): unopened flowers; (**b**): opened flowers).

**Figure 2 jof-10-00145-f002:**
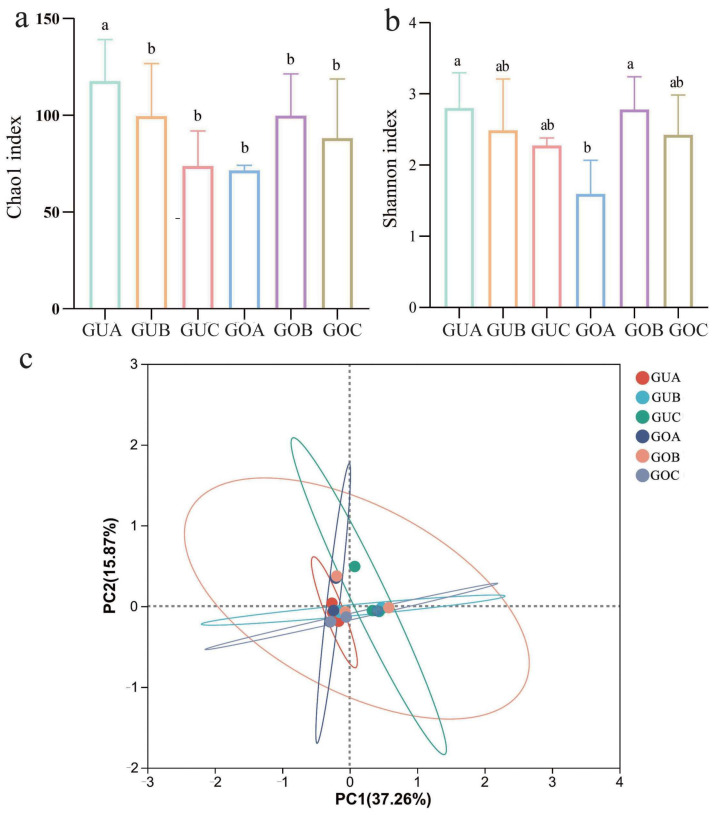
Analysis of fungal diversity and community in the flower of *C*. *praecox* in different blooming stages and under different circadian rhythms. (**a**) Chao1 index; (**b**) Shannon index; (**c**) fungal community. GUA: unopened flowers at 7:00 a.m.; GUB: unopened flowers at 1:00 p.m.; GUC: unopened flowers at 7:00 p.m.; GOA: opened flowers at 7:00 a.m.; GOB: opened flowers at 1:00 p.m.; GOC: opened flowers at 7:00 p.m.. Different letters in each column indicate significant differences (*p* < 0.05).

**Figure 3 jof-10-00145-f003:**
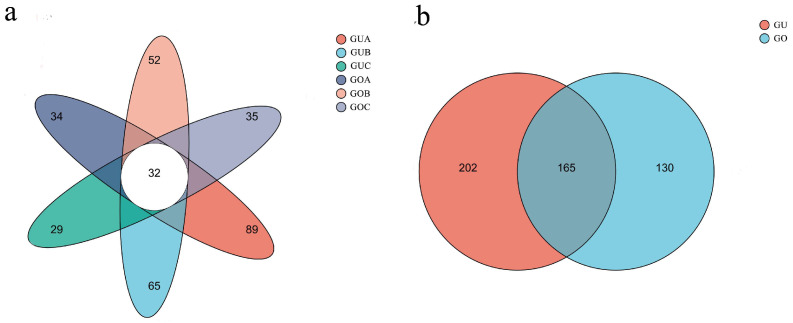
Venn analysis for the OTU distribution of endophytic fungi. (**a**,**b**) indicate the different blooming stages and circadian rhythms, respectively. GUA: unopened flowers at 7:00 a.m.; GUB: unopened flowers at 1:00 p.m.; GUC: unopened flowers at 7:00 p.m.; GOA: opened flowers at 7:00 a.m.; GOB: opened flowers at 1:00 p.m.; GOC: opened flowers at 7:00 p.m..

**Figure 4 jof-10-00145-f004:**
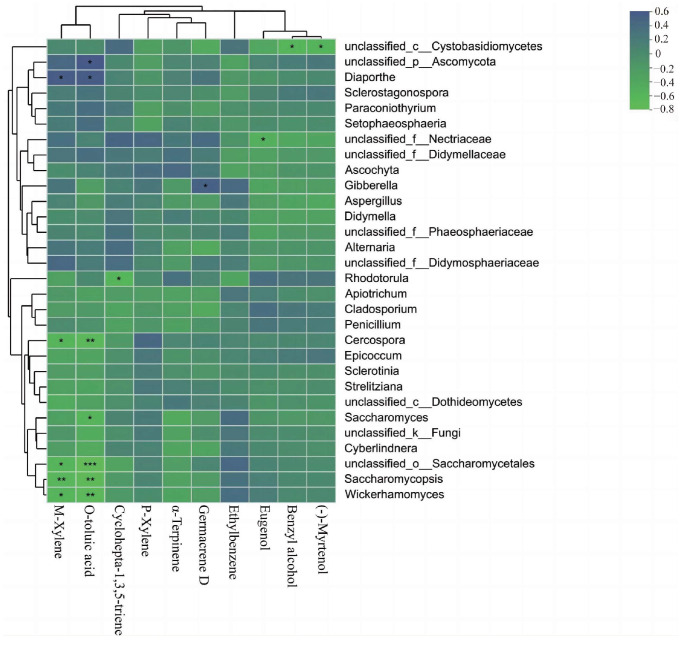
Correlation analysis for the relationship between the main floral compounds and endophytic fungi in *C. praecox*. ***, ** and * indicates a significant correlation at the 0.001, 0.01 and 0.05 levels, respectively.

**Table 1 jof-10-00145-t001:** Relative abundance of the top five fungi at different levels of taxonomy in the six samples.

Taxonomy Levels	Top Five	GUA	GUB	GUC	GOA	GOB	GOC
Phylum	Ascomycota	96.57%	90.84%	87.70%	98.40%	85.16%	82.32%
	Basidiomycota	2.88%	7.03%	8.85%	1.55%	13.60%	13.60%
	unclassified_k__Fungi	0.29%	1.95%	3.07%	0.04%	0.86%	3.31%
Class	Dothideomycetes	46.20%	37.16%	44.57%	44.03%	36.64%	19.90%
	unclassified_p__Ascomycota	35.23%	20.61%	7.43%	50.52%	26.31%	37.55%
	Eurotiomycetes	8.30%	7.34%	8.39%	2.41%	8.81%	4.32%
	Sordariomycetes	5.73%	16.97%	4.06%	0.89%	1.30%	14.92%
	Tremellomycetes	1.94%	5.66%	6.93%	1.00%	10.83%	11.45%
Order	Pleosporales	33.37%	31.01%	18.37%	40.96%	50.53%	30.30%
	unclassified_p__Ascomycota	46.20%	20.61%	41.66%	7.43%	41.98%	26.13%
	Eurotiales	7.81%	4.45%	16.93%	5.94%	1.30%	7.42%
	Hypocreales	4.51%	16.82%	1.57%	1.99%	0.49%	0.29%
	Trichosporonales	1.73%	5.49%	1.63%	5.06%	0.73%	9.67%
Family	unclassified_p__Ascomycota	46.20%	20.61%	7.43%	50.52%	26.13%	37.55%
	Didymellaceae	18.86%	23.55%	25.44%	28.41%	19.57%	9.13%
	Phaeosphaeriaceae	10.68%	6.00%	14.21%	4.20%	7.45%	4.29%
	Aspergillaceae	7.80%	4.45%	5.94%	1.29%	7.20%	2.45%
	Nectriaceae	4.49%	16.46%	1.62%	0.47%	1.63%	13.67%
Genus	unclassified_p__Ascomycota	46.20%	20.61%	24.58%	50.52%	26.13%	37.55%
	Didymella	17.39%	19.57%	7.43%	26.91%	14.77%	8.70%
	unclassified_f__Phaeosphaeriaceae	9.88%	4.61%	12.15%	2.84%	4.05%	3.57%
	Aspergillus	7.29%	5.49%	5.06%	1.19%	6.32%	1.63%
	Gibberalla	3.86%	13.87%	1.28%	0.73%	1.63%	13.28%
Species	unclassified_p__Ascomycota	46.20%	20.61%	24.58%	50.52%	26.13%	37.56%
	unclassified_g__Didymella	17.39%	19.56%	7.43%	26.91%	14.77%	8.70%
	unclassified_f__Phaeosphaeriaceae	9.88%	4.61%	12.15%	2.84%	4.05%	3.57%
	Aspergillus amstelodami	6.49%	2.08%	2.12%	0.84%	1.41%	10.71%
	unclassified_g__Gibberella	3.61%	13.13%	1.05%	8.99%	9.67%	13.28%

GUA: unopened flowers at 7:00 a.m.; GUB: unopened flowers at 1:00 p.m.; GUC: unopened flowers at 7:00 p.m.; GOA: opened flowers at 7:00 a.m.; GOB: opened flowers at 1:00 p.m.; GOC: opened flowers at 7:00 p.m..

## Data Availability

The sequences have been deposited in the National Center for Biotechnology Information Sequence Read Archive under Accession no. PRJNA875038 (ITS data).

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
