# Peer review of "The Relationship between Endophytic Fungi of Chimonanthus praecox and Volatile Metabolites under Different Circadian Rhythms and Blooming Stages"

_jof, 2024, doi:10.3390/jof10020145_

Round 1
Reviewer 1 Report
Comments and Suggestions for Authors
The authors submitted a paper focused on the intriguing topic of the dynamics in symbiotic communities of fungi and plants. This topic seems to be topical and has been actively studied in recent years. The work was based on the latest molecular methods for the study of the hidden diversity. The authors have obtained valuable data on the dynamics of endophytic fungi and the factors determining it. Additional interest of the work is given by the attempt to compare the composition of the fungal community and the synthesis of secondary metabolites. This is of interest both for the study of plant-fungus interactions and may find practical application. Nevertheless, the work has a number of essential drawbacks that need to be corrected.
- Authors conclude that diurnal cycles have an effect on the diversity of endophytic fungi. This is an important and nontrivial conclusion because, at first glance, a day seems like a very short period for community change. The authors should pay more attention to the consideration of such cases in the introduction and discussion. The discussion would also do well to consider the mechanisms of such changes in more detail.
- Using PCoA based on the Bray-Curtis index, the authors write that there is no difference between the groups. How does this relate to the conclusions about the influence of the factors studied? Perhaps these differences should be examined in more detail and using other methods (more components, PERMANOVA, MDS, etc.).
- The authors analyze a volatile compounds. How were the data obtained?
- The statistical methods applied in the paper should be described in more detail.
- The illustrative material needs improvement. The scatter plots need to use not only colors but also the shape of the points. The heat map can be made more contrasting.
The paper can be published after major revision.
Author Response
Dear editors and reviewers,
Firstly, we would like to express our sincere appreciation for your effort on the manuscript! The manuscript has been extensively revised as the reviewer suggests. the details are listed below. Moreover, the language in the article has been polished under the MDPI company. Finally, thank you again for your positive comments and valuable suggestions, which helped us to improve the manuscript to a better scientific level.
Reviewer #1:
1. Authors conclude that diurnal cycles have an effect on the diversity of endophytic fungi. This is an important and nontrivial conclusion because, at first glance, a day seems like a very short period for community change. The authors should pay more attention to the consideration of such cases in the introduction and discussion. The discussion would also do well to consider the mechanisms of such changes in more detail:
Thank you for your suggestions. Changes in the diversity of endophytic fungi in response to circadian rhythms is a novel point found in our content. No reports have been found on this. We have added this sentence in lines 279-286.: As insect pollinators act rhythmically, the variation of floral scent with rhythmically are associated with the activity of corresponding pollinators. This leads us to speculate a potential close relationship between plant endophytic fungi and their secondary metabolites.
2. Using PCoA based on the Bray-Curtis index, the authors write that there is no difference between the groups. How does this relate to the conclusions about the influence of the factors studied? Perhaps these differences should be examined in more detail and using other methods (more components, PERMANOVA, MDS, etc.).
It was indicated that the fungal communities are similar at different blooming stages and circadian rhythms, while the fungal abundance differed. We also did a PERMANOVA (R2=0.22, P>0.05) and NMDS analysis resulted in consistent results.
3. The authors analyze a volatile compounds. How were the data obtained?
Based on our previous research (reference 37), a specific determination method has been added in lines 122-131.
4. The statistical methods applied in the paper should be described in more detail.
Thank you very much for your advice! It has been added in lines 163-169.
5. The illustrative material needs improvement. The scatter plots need to use not only colors but also the shape of the points. The heat map can be made more contrasting
Thanks for your suggestion, we have modified the scatter plot, and made it clearer.
Reviewer 2 Report
Comments and Suggestions for Authors
The Authors used NGS to investigate the fungal community in Chimonanthus praecox at different times. They also measured the floral volatiles and completed correlation analysis.
The manuscript is interesting, but the information needs to be presented in a clearer manner. Please see the attached file for comments, suggestions, and line points.
I think the idea of correlating the volatile compounds to species is interesting (although correlation is not causation). However, the introduction focuses on fungi contributing to the volatile production only. At quick glance, it looked like some of these chemicals (volatile aromatics) are likely antifungal as well or potentially used as food sources by some fungi. The authors should be clearer throughout the manuscript about this aspect of manuscript. I also believe the authors should discuss other potential reasons for their results.
For the Supplemental files:
1) All tables and figures need to be standalone, so please define all abbreviations.
2) Figure is spelled incorrectly.
3) Figure S2 is based on genera so everything that is unclassified should be placed into one bin, not spread out across phylum, class, order, or family. By default, they are unclassified.

The English grammar and structure need some improvements in places (see attached file for comments).
Author Response
Dear editors and reviewers,
Firstly, we would like to express our sincere appreciation for your effort on the manuscript! The manuscript has been extensively revised as the reviewer suggestion. the details are listed below. Moreover, the language in the article has been polished under the MDPI company. Finally, thank you again for your positive comments and valuable suggestions, which helped us to improve the manuscript to a better scientific level.
Reviewer #2:
I think the idea of correlating the volatile compounds to species is interesting (although correlation is not causation). However, the introduction focuses on fungi contributing to the volatile production only. At quick glance, it looked like some of these chemicals (volatile aromatics) are likely antifungal as well or potentially used as food sources by some fungi. The authors should be clearer throughout the manuscript about this aspect of manuscript. I also believe the authors should discuss other potential reasons for their results.
Thank you very much for your advice! I have added the corresponding content in line 297-301 of the discussion. “These aromatic compounds serve as food for endophytic fungi and resist the toxicity of the compounds themselves”.
1. All tables and figures need to be standalone, so please define all abbreviations.
We have modified it.
2. Figure is spelled incorrectly.
I've corrected it.
3. Figure S2 is based on genera so everything that is unclassified should be placed into one bin, not spread out across phylum, class, order, or family. By default, they are unclassified.
We have categorized at the genus level on the data platform, and all unclassified strains can determined in the library for the family, phylum, or order identification.
Round 2
Reviewer 1 Report
I would like to thank the author for the informative answer. The answer is satisfying.
I wish the authors great success in their future research.
Overall, the work looks complete and ready for publication.
Author Response
Dear editors and reviewers,
Thank you again for your positive comments and valuable suggestions. We have revised the manuscript again.
Reviewer 2 Report
The manuscript has greatly improved and most of my concerns have been addressed. The only concern is related to the interpretation related to the unknown OTUs. Based on Figure S2, which is the breakdown of the community at the genus level, there are many non-genera groups in the figure. What I mean is that it appears that all unclassified OTUs at the Kingdom, Class, Order, or Family level have been grouped together to indicate proportion. However, it is very likely that each of these unclassified groups contain more than one genus. Therefore, because they are unclassified and grouped together, they cannot be discussed as a genus, and the section that discusses this in the manuscript needs to talk only about the identified OTUs.
There are some detailed comments in the attached document.
1) In general, make sure to italicize all genera
2) Do not italicize higher taxonomic rank
3) Figure S2 needs to be corrected to combine all unclassified into one single group (unclassified group) if the authors keep it, and they should not discuss this group in terms of being the most abundant because they likely represent multiple taxa.

Author Response
Dear editors and reviewers,
Thank you again for your positive comments and valuable suggestions. We have revised the manuscript again, the details are as follows:
Reviewer #2:
- In general, make sure to italicize all genera
Thank you very much for your amendment, we have corrected it in the manuscript!
- Do not italicize higher taxonomic rank
Thank you. We have corrected it!
3) Figure S2 needs to be corrected to combine all unclassified into one single group (unclassified group) if the authors keep it, and they should not discuss this group in terms of being the most abundant because they likely represent multiple taxa.
Thank you for your suggestions. We have redrawn Figure S2, and put the unclassified fungi in a group as you suggested.